Is grazing exclusion effective in restoring vegetation in degraded alpine grasslands in Tibet, China?

Yan Yan
Lu Xuyang xylu@imde.ac.cn
Key Laboratory of Mountain Surface Processes and Ecological Regulation, Institute of Mountain Hazards & Environment, Chinese Academy of Sciences , Chengdu , China
Sandhu Harpinder
Electronic publication date: 2015 Jun 16
Publication date: 2015
Volume: 3
Electronic Location ID: e1020
Received 2015 Mar 26; Accepted 2015 May 24
Copyright: © 2015 Yan and Lu
Copyright year: 2015
Copyright holder: Yan and Lu
License: This is an open access article distributed under the terms of the Creative Commons Attribution License, which permits unrestricted use, distribution, reproduction and adaptation in any medium and for any purpose provided that it is properly attributed. For attribution, the original author(s), title, publication source (PeerJ) and either DOI or URL of the article must be cited.
License URL: https://creativecommons.org/licenses/by/4.0/

Keywords: Grazing exclusion, Alpine grassland, Grassland degradation, Community characteristics, Biomass, Tibet

Funding: National Natural Science Foundation of China 41201053 135 Strategic Programs of the Institute of Mountain Hazards and Environment sds-135-1203-03 Action Plan of the Chinese Academy of Sciences for Western Development KZCX2-XB3-08 This study was funded by the National Natural Science Foundation of China (41201053), the 135 Strategic Programs of the Institute of Mountain Hazards and Environment (sds-135-1203-03), and the Action Plan of the Chinese Academy of Sciences (CAS) for Western Development (KZCX2-XB3-08). The funders had no role in study design, data collection and analysis, decision to publish, or preparation of the manuscript.

==============================
Overgrazing is considered one of the key disturbance factors that results in alpine grassland degradation in Tibet. Grazing exclusion by fencing has been widely used as an approach to restore degraded grasslands in Tibet since 2004. Is the grazing exclusion management strategy effective for the vegetation restoration of degraded alpine grasslands? Three alpine grassland types were selected in Tibet to investigate the effect of grazing exclusion on plant community structure and biomass. Our results showed that species biodiversity indicators, including the Pielou evenness index, the Shannon–Wiener diversity index, and the Simpson dominance index, did not significantly change under grazing exclusion conditions. In contrast, the total vegetation cover, the mean vegetation height of the community, and the aboveground biomass were significantly higher in the grazing exclusion grasslands than in the free grazed grasslands. These results indicated that grazing exclusion is an effective measure for maintaining community stability and improving aboveground vegetation growth in alpine grasslands. However, the statistical analysis showed that the growing season precipitation (GSP) plays a more important role than grazing exclusion in which influence on vegetation in alpine grasslands. In addition, because the results of the present study come from short term (6–8 years) grazing exclusion, it is still uncertain whether these improvements will be continuable if grazing exclusion is continuously implemented. Therefore, the assessments of the ecological effects of the grazing exclusion management strategy on degraded alpine grasslands in Tibet still need long term continued research.

Introduction

Tibet is an important ecological security shelter zone that acts as an important reservoir for water, regulating climate change and water resources in China and eastern Asia (Sun et al., 2012). Alpine grasslands are the most dominant ecosystems over all of Tibet, covering more than 70% of the whole plateau’s area and representing much of the land area on the Eurasian continent (Wang et al., 2002). Alpine grasslands in this area are grazed by indigenous herbivores, such as yak and Tibetan sheep. These ecosystems have traditionally served as the principal pastures for Tibetan communities and are regarded as one of the major pastoral production bases in China (Wen et al., 2013a). Alpine grasslands also provide ecosystem functions and services, such as carbon sequestration, biodiversity conservation, and soil and water protection, and are also of great importance for Tibetan culture and the maintenance of Tibetan traditions (Wen et al., 2013b; Shang et al., 2014).

Alpine grasslands in Tibet have been regionally degrading, even desertifying since the 1980s. For instance, from 1981 to 2004 in northern Tibet, which is the main extent of alpine grassland distribution and an important livestock production centre in Tibet, degraded alpine grasslands accounted for 50.8% of the total grassland area, and severely and extremely severely degraded grasslands accounted for 8.0% and 1.7%, respectively (Gao et al., 2010). Grassland degradation may be due to a combination of global climate change, rapidly increasing grazing pressure, rodent damage and other factors (Chen et al., 2014). Nevertheless, overgrazing, which caused by an increase in the population of humans and domestic livestock in Tibet, is widely considered the primary cause of grassland degradation. Overgrazing may result in significant changes to the composition and structure of the plant community including significant decreases in the regenerative ability of the grasslands, decreases in biomass, decreases in the amount of nutrients returned to the soil as litter, and eventually cause grassland degradation (Zhou et al., 2005). Additionally, overgrazing causes an increase in potential evapotranspiration, thereby promoting the warming of local climate and further accelerating alpine grassland degradation processes (Du et al., 2004). Under conditions of overgrazing by livestock, the succession of degraded grasslands can become a vicious circle: overgrazing causes grassland degradation, which facilitates rodent infestation, which further degrades grasslands (Kang et al., 2007).

In an attempt to alleviate the problem of grassland degradation in Tibet, China’s state and local authorities initiated a program in 2004 called ‘retire livestock and restore pastures’ (Fig. 1). As part of this campaign, grazing exclusion by fencing has been widely used as an approach to the restoration grasslands (Wei et al., 2012). Grazing exclusion is an effective grassland management practice that aimed to prevent grassland degradation and retain grassland ecosystem function throughout the world in recent decades (Mata-González et al., 2007; Mofidi et al., 2013). This management strategy is expected to restore vegetation and enhance rangeland health in overgrazed and degraded grasslands characterized by low productivity, low vegetation cover and low biomass in Tibet. This campaign has been in progress for more than ten years, which brings to light the question: is this program successful in the restoration of degraded alpine grasslands? This question has attracted great attention in recent years and has inspired a large number of studies on the effect of grazing exclusion on the alpine grasslands (Wei et al., 2012; Wu et al., 2012; Shi et al., 2013).

Figure 1 The ‘retire livestock and restore pastures’ program in Tibet.

(A) A grazing exclusion sign set by the government, (B) the fence-line contrast between the fenced and grazed grassland.

Nevertheless, research results with regard to the effect of grazing exclusion on plant biomass and biodiversity were not consistent. For instance, grazing exclusion could result in the improvement of grass cover, species biodiversity and biomass due to the absence of grazing in some degraded grassland ecosystems (Mata-González et al., 2007; Mofidi et al., 2013). However, grazing exclusion may induce a decrease of species richness and biodiversity by the replacement of species that are highly adapted to grazing by strongly dominant competitors that increase in abundance due to grazing cessation, such as certain graminoids (Mayer et al., 2009; Shi et al., 2013). The lack of a consistent response of vegetation to grazing exclusion has been attributed to a broad range of factors that determine whether and how herbivores affect plant communities, include the time of grazing exclusion (Mayer et al., 2009), growing season precipitation (Wu et al., 2012), productivity (Schultz, Morgan & Lunt, 2011), climatic conditions (Jing, Cheng & Chen, 2013) and so on. Therefore, specific studies are crucial to that ecosystems are properly managed and that conservation goals are achieved. In the Tibetan Plateau, although numerous studies exploring the effect of grazing exclusion on alpine grassland ecosystem structure and function have been published in recent years (Wu et al., 2009; Shang et al., 2013; Li et al., 2014; Luan et al., 2014; Zhang et al., 2015), the majority of studies focus on a single alpine grassland ecosystem type and in one experimental site. Little studies focus on the assessment of the effects of grazing exclusion on alpine grassland ecosystem on regional scale (Wu et al., 2013; Wu et al., 2014).

To gain a better understanding of the restoration and management of degraded grasslands in Tibet, studies are needed to investigate alpine grassland vegetation growth and community composition dynamics. Thus, the aim of this study was to investigate the effects of excluding grazing herbivores through fencing on high-altitude alpine grasslands in Tibet, and to assess whether fencing can be used as an effective grassland management tool to restore vegetation in degraded alpine grasslands. Three alpine grassland types and nine counties, which represent the main natural alpine grassland distribution in Tibet, were selected as sampled sites according to the time and range of grazing exclusion. We hypothesized that in the absence of grazing, the vegetation cover, height, the above- and below-ground biomass, species richness, diversity would improve due to the absence of disturbance from herbivorous livestock. In addition, based on different plant species diversity and community structure, vegetation productivity and cover, and environmental conditions, we further hypothesized that vegetation biomass and biodiversity responses to the absence of grazing would differ among different alpine grassland types.

Methods

Study area

Tibet is located between 26°50′ and 36°29′N and 78°15′ and 99°07′E and covers a total area of more than 1.2 million km2, which is approximately one-eighth of the total area of China (Fig. 2). The main portion of the Qinghai-Tibetan Plateau lies at an average altitude of 4,500 m; it is geomorphologically unique in the world. Because of its extensive territory and highly dissected topography, the region has a diverse range of climate and vegetation zones. The solar annual radiation is strong and varies between 140 and 190 kcal cm−2 in different parts of the region. Annual sunshine tends to increase from the east to the west and ranges from 1,800 to 3,200 h. The average annual temperature is rather low, with a large diurnal range, and varies from 18 °C to −4 °C; the average temperature in January varies from 10 °C to −16 °C; the average temperature in July varies from 24 °C to 8 °C, and decreases gradually from the southeast to the northwest. The average annual precipitation is less than 1,000 mm in most areas of Tibet, and reaching up to 2,817 mm in the east and dropping down to approximately 70 mm in the west (Zou et al., 2002; Dai et al., 2011).

Figure 2 Location of sampling sites of alpine grasslands in Tibet.

According to the first national survey of Chinese grassland resources, Tibet ranks first among all Chinese provinces and autonomous regions in the diversity of its grassland ecosystems, comprising 17 types of grassland based on the classification system used for the whole country (Gai et al., 2009). Among all grassland types, alpine steppe is the most common grassland type in Tibet; it is composed of drought tolerant perennial herb or small shrubs under cold and arid and semiarid climate conditions, and represents approximately 38.9% of the total Tibetan grassland area. Alpine meadow is the second largest grassland type, and is composed of perennial mesic and mesoxeric herbs under cold and wet climate conditions, occupying approximately 31.3% of the total grassland area of Tibet. Alpine desert steppe occupies approximately 10.7% of the total grassland area, and is composed by xeric small shrubs and small grasses under cold and arid climate conditions; it is a transitional type of alpine grassland from the steppe to the desert in Tibet (Land Management Bureau of Tibet, 1994).

Survey design, sampling, and data collection

Since the ‘retire livestock and restore pastures’ ecological engineering program started in 2004, more than 2.4 × 106 ha of alpine grasslands in Tibet have been fenced to exclude livestock grazing. Nine counties in Tibet, in which the extent of fenced area was relatively larger, were selected to investigate the effect of grazing exclusion on plant community composition and biomass in alpine grasslands (Fig. 2). These nine counties represented three of the main natural grassland vegetation types in Tibet, including alpine meadow, alpine steppe and alpine desert steppe (Table 1). In these counties, areas which were fenced over the years of 2005–2007 were chosen as sampling sites in the present study. Since fencing establishment, the fenced grassland has completely been excluded from livestock grazing, while the surrounding grassland continued conventional grazing by yak and sheep around the year. No accurate information, such as timing, intensity, and frequency, on grazing activities and pasture management in the open free grazing grassland, but the actual averaged stocking rate ranges were approximately from 0.16 sheep units hm−2 of the westernmost county to 2.05 sheep units hm−2 in the easternmost county for the study region (Wu et al., 2014). In addition, no specific permits were required for the described field studies and the field studies did not involve protected animals or plants. The enclosed areas were defined as grazing exclusion (GE) plot and the areas outside of fencing were defined as free grazing (FG) plot. Field surveys were conducted during late July to mid-August in 2013; three pairs (fenced versus free grazed) of plots in each site were chosen and surveyed.

Table 1 Description of the sampling sites of alpine grasslands in Tibet.

Location	Longitude (E)	Latitude (N)	Altitude (m)	Grassland type	Dominant species	
Damxung	91°14′56″	30°36′08″	4,407	Alpine meadow	Kobresia pygmaea C. B. Clarke	
Nagqu	92°09′11″	31°16′30″	4,458	Alpine meadow	Kobresia humilis	
Nierong	92°16′49″	32°07′48″	4,614	Alpine meadow	Kobresia littledalei C. B. Clarke	
Ando	91°38′28″	32°15′37″	4,696	Alpine meadow	Kobresia pygmaea C. B. Clarke	
Baingoin	92°09′11″	31°16′30″	4,632	Alpine steppe	Carex moorcroftii Falc. Ex Boott	
Nima	87°24′57″	31°48′27″	4,550	Alpine steppe	Stipa purpurea	
Coqen	85°09′09″	31°01′58″	4,687	Alpine steppe	Stipa purpurea	
Ngamring	86°37′52″	29°38′38″	4,583	Alpine steppe	Carex moorcroftii Falc. Ex Boott	
Gêrzê	84°49′34″	31°59′25″	4,591	Alpine desert steppe	Stipa purpurea	

At each sample plot, three pairs of 0.5 m × 0.5 m quadrats at each GE and FG treatment sample plots were laid out collinearly at intervals of approximately 20 m. All species within each quadrat were identified and their coverage, density, frequency and natural height were measured. The frequency counts were made by dividing the 0.5 m × 0.5 m frame into 10 cm × 10 cm cells. Within each cell, presence/absence species data were recorded. These data were summed up to calculate frequencies per quadrat (1–100%). The geographical coordinates, elevation and vegetation types for each site were also recorded, and the picture of each quadrat was taken using a digital camera to calculate community cover. Aboveground and belowground plant components were harvested. Aboveground plant parts in the sample quadrat were clipped to the soil surface with scissors and belowground plant parts in the sample quadrat were directly acquired by excavation, including both live roots and dead roots of all plant species. After sun-drying of plant samples in the field, they were brought to laboratory and oven-dried at 65 °C for 72 h to determine biomass.

Monthly meteorological datasets with spatial resolutions of 0.5° from 2005 to 2013, which generated by Thin Plate Spline (TPS) method using ANUSPLIN software (ERSI, Redlands, California, USA) and the data sources include monthly mean temperature and monthly precipitation data from more than 2,400 well distributed climate stations across China, were derived from the China Meteorological Data Sharing Service System (http://cdc.nmic.cn). The average growing season (from May to September) temperature (GST) and growing season precipitation (GSP) from 2005 to 2013 matched with nine sites’ locations were extracted from these meteorological raster surfaces in ArcGIS 10.0 (ERSI, Redlands, California, USA) for further analyses.

Plant community characteristics

Total cover, community vegetation height, and the Simpson index, Shannon index, and Pielou index were used to describe the plant community characteristics of alpine grassland ecosystems. The vegetation total cover was acquired from pictures of each quadrat by using CAN-EYE software (INRA-UAPV, France) and the vegetation height of the community was directly measured as the height of the dominant vegetation within each quadrat. To reveal the variation in community composition characteristics in grazing exclusion process, the Pielou evenness index (E), Shannon diversity index (H), and Simpson dominance index (D) were used to indicate plant community biodiversity changes. The Pielou evenness index reflects allocation information and species composition. The Shannon diversity index, which ranging in theory from 0 to infinity and incorporating both species richness and evenness aspects together, increases as the number of species increases and as individuals are evenly distributed among species. The Simpson dominance index gives the probability of two randomly chosen individuals drawn from a population belonging to the same species, a higher value also indicates a higher diversity.

The following formulas were used to calculate Pielou evenness index (E), Shannon–Wiener diversity index (H), and Simpson dominance index (D): E=−∑PilnPi/lnS

H=−∑PilnPi

D=1−∑Pi2

where Pi = ni/T, ni is the count of each plant species i in a quadrat, T is the total count of all plant species in a quadrat, in that context, Pi is the relative probability of finding the species i in a quadrat. S is the total observed number of species in a quadrat.

Statistical analysis

A paired difference t-test was conducted to test differences in the examined parameters between fenced and grazed plots within each grassland type. Analysis of covariance (ANCOVA) by the general linear model (GLM) was employed to evaluate the effects of grazing exclusion treatment and climatic factors on the plant community and biomass indices in Tibet. In the ANCOVA analysis, fixed factor was alpine grassland grazing treatments (FG and GE), while the covariates were GST and GSP. The two covariates GST and GSP that were used to fit the linear ANCOVA models were not highly interacted with the fixed factor (P > 0.05). Pearson correlation analysis was used to test the relationships between different plant community composition and biomass indices. The least significant difference test was used to compare the means at P < 0.05. All statistical analyses were performed using IBM SPSS Statistics 19 software (SPSS/IBM, Chicago, IL, USA).

Results

Plant community characteristics

Changes in the selected plant community characteristics are shown in Table 2. Compared to the FG plots, vegetation total cover of alpine grassland (alpine meadow + alpine steppe + alpine desert steppe) was 8.83% higher (P < 0.05) and the community vegetation height was 2.65 cm higher (P < 0.05) in the GE plots. However, grazing exclusion did not significantly affect the biodiversity of alpine grassland (P > 0.05), although the Simpson index, Shannon index, and Pielou index in GE plots were 0.03, 0.07, and 0.01 lower than those indices in FG plots, respectively. Among three alpine grassland types, there was no significant difference in the Simpson index, Shannon index, or Pielou index between FG and GE plots in alpine meadow, alpine steppe or alpine desert steppe. Nevertheless, significant difference in total cover was found in alpine steppe and in community vegetation height was found in both alpine meadow and alpine steppe (P < 0.05). The results from ANCOVA analysis demonstrated that grazing exclusion had a significant effect on vegetation cover and vegetation height, but did not affect biodiversity indices, including D, H, and E. For growing season climate factors, GST had a significant effect on D and H, whereas GSP had a significant effect on most plant community characteristic indices (Table 3).

Table 2 Plant community characteristics and biomass.

Statistical comparison of overall mean values of plant community characteristics and biomass indices ± standard error (S.E.) in alpine grassland by using paired difference t-test (α = 0.05) between free grazing (FG) plots and grazing exclusion (GE) plots. P-values below 0.05 are in bold.

Indices	Site	Alpine meadow	Alpine steppe	Alpine desert steppe	Alpine grassland	
Total cover (TC)	FG (%)	61.32 ± 5.29	14.96 ± 1.36	6.74 ± 1.43	34.65 ± 5.27	
GE (%)	70.07 ± 3.54	25.59 ± 3.46	8.74 ± 0.46	43.48 ± 5.23	
Vegetation height (VH)	FG (cm)	2.62 ± 0.56	5.39 ± 0.30	5.77 ± 0.29	4.20 ± 0.39	
GE (cm)	5.63 ± 1.02	8.13 ± 1.11	6.66 ± 0.19	6.85 ± 0.70	
Simpson index (D)	FG	0.37 ± 0.05	0.67 ± 0.02	0.43 ± 0.05	0.51 ± 0.04	
GE	0.35 ± 0.07	0.66 ± 0.02	0.23 ± 0.07	0.48 ± 0.05	
Shannon index (H)	FG	0.80 ± 0.11	1.32 ± 0.08	0.71 ± 0.10	1.02 ± 0.08	
GE	0.74 ± 0.13	1.31 ± 0.05	0.38 ± 0.10	0.95 ± 0.09	
Pielou index (E)	FG	0.42 ± 0.05	0.76 ± 0.02	0.65 ± 0.09	0.60 ± 0.04	
GE	0.43 ± 0.07	0.76 ± 0.04	0.55 ± 0.14	0.59 ± 0.05	
Aboveground biomass (AB)	FG (g m−2)	70.86 ± 10.82	33.95 ± 4.86	17.15 ± 5.39	48.49 ± 6.56	
GE (g m−2)	80.62 ± 10.94	55.39 ± 5.78	31.23 ± 3.69	63.92 ± 6.28	
Belowground biomass (BB)	FG (g m−2)	628.27 ± 240.67	203.30 ± 26.49	218.33 ± 46.71	393.85 ± 112.84	
GE (g m−2)	1,010.86 ± 265.88	272.80 ± 33.02	232.67 ± 36.18	596.37 ± 137.11	
Total biomass (TB)	FG (g m−2)	699.13 ± 241.52	237.25 ± 28.49	235.48 ± 51.96	442.34 ± 114.77	
GE (g m−2)	1,091.48 ± 263.27	328.19 ± 37.24	263.89 ± 35.96	660.28 ± 137.98	

Table 3 Effects of grazing exclusion and climate factors.

Results from analysis of covariance (ANCOVA) by the general linear model (GLM) showing F values and P values of plant community characteristics and biomass indices, which the fixed factor was grazing treatments (free grazing and grazing exclusion) and the covariates were growing season temperature (GST) and growing season precipitation (GSP). P-values below 0.05 are in bold.

Indices	Grazing exclusion	GST	GSP	
	F-value	P-value	F-value	P-value	F-value	P-value	
Total cover (TC)	22.82	<0.001	0.60	0.443	52.27	<0.001	
Vegetation height (VH)	11.35	0.001	2.44	0.124	0.32	0.572	
Simpson index (D)	0.61	0.440	8.24	0.006	10.27	0.002	
Shannon index (H)	0.62	0.435	13.81	0.001	6.99	0.011	
Pielou index (E)	0.09	0.771	2.51	0.120	20.86	<0.001	
Aboveground biomass (AB)	4.07	0.049	0.01	0.950	20.69	<0.001	
Belowground biomass (BB)	1.58	0.215	0.00	0.982	11.66	0.001	
Total biomass (TB)	1.84	0.181	0.00	0.980	13.28	0.001	

Aboveground and belowground biomass

Grazing exclusion had a significant effect on aboveground biomass of alpine grasslands, the mean value of aboveground biomass of the GE plots were 15.43 g cm−2, higher than that of FG plots (P < 0.05). However, grazing exclusion had no significant effect on belowground biomass and total biomass (aboveground biomass + belowground biomass) (P > 0.05, Table 2). Among the three alpine grassland types, for alpine meadow, there were no significant changes in biomass features 6–8 years after fencing, including aboveground, belowground, and total biomass. For alpine steppe, the aboveground, belowground, and total biomasses were all significantly increased due to grazing exclusion (P < 0.05). Moreover, grazing exclusion led to a significant increase in the aboveground biomass in alpine desert steppe (P < 0.05, Table 2). Statistical analyses from ANCOVA showed that grazing exclusion had a significant effect on aboveground biomass of alpine grassland ecosystems in Tibet (P < 0.01, Table 3). The effects of climate factors on biomass were differ between GST and GSP, which GSP had a significant effect on all biomass indices, but GST had no any effect on biomass of alpine grasslands (Table 3).

Relationship among community characteristics and biomass indices

Correlation analyses showed that total cover was negatively correlated with D, H and E (P < 0.01), and significant positively correlated with aboveground, belowground, and total biomass (P < 0.01). However, community vegetation height was positively correlated with D and E (P < 0.01), and no correlations were found between the community vegetation height and any of the biomass parameters (P > 0.05). The community biodiversity indices D, H, and E were all positively correlated with each other (P < 0.01). In addition, the total biomass was positively correlated with the belowground biomass (P < 0.01; Table 4).

Table 4 Correlation relationships among plant community characteristics and biomass indices.

Pearson’s correlation coefficients among plant community characteristics and biomass indices of alpine grasslands in Tibet, and their significance levels.

Indices	TC	VH	D	H	E	AB	BB	
VH	−0.07							
D	−0.49**	0.29*						
H	−0.40**	0.23	0.97**					
E	−0.64**	0.32*	0.92**	0.84**				
AB	0.72**	0.18	−0.21	−0.17	−0.34*			
BB	0.47**	−0.05	−0.27*	−0.24	−0.29*	0.21		
TB	0.50**	−0.04	−0.28*	−0.24	−0.30*	0.26	0.99**	
Notes.

* P < 0.05.

** P < 0.01.

TC Total coverage

VH Vegetation height

D Simpson index

H Shannon index

E Pielou index

AB Aboveground biomass

BB Belowground biomass

TB Total biomass

Discussion

Overgrazing due to sharp growth of the human population and of food demand in recent years is a major cause of grassland degradation on the Tibetan Plateau (Wei et al., 2012; Shang et al., 2014). Grassland degradation has significantly altered species composition and decreased productivity in the region (Zhou et al., 2006; Ma, Zhou & Du, 2013). The exclusion of livestock through the use of mesh fencing to create large-scale enclosures has become a common grassland management strategy for restoring degraded grasslands of the Tibetan Plateau in recent decades (Wu et al., 2009; Shi et al., 2013). Is grazing exclusion an effective policy to restore vegetation in degraded alpine grassland in Tibet? In the present study, three alpine grassland types and nine counties were selected as sampled sites according to the time and range of grazing exclusion to investigate the effects of grazing exclusion by fencing on plant community characteristics and biomass in degraded alpine grasslands.

Impacts of grazing exclusion on community characteristics

Vegetation cover is an important index for measuring the protective function of vegetation to the ground. Our study shows that continuous grazing exclusion resulted in a significant increase in the total vegetation cover of alpine grasslands (Table 2). This result is consistent with previous reports, supporting the conclusion that the exclusion of grazing livestock in the degraded alpine grasslands of the Tibetan Plateau exerts a strong effect on ecosystem dynamics by increasing vegetative cover (Wu et al., 2009; Shang et al., 2013). The mean vegetation height of community in GE plots was 6.85 cm, which was approximately 1.63 times that in the FG plots (Table 2). Similar results have also been reported in other studies of alpine grasslands in Tibet (Deléglise, Loucougaray & Alard, 2011; Shang et al., 2013). Increased vegetation cover and height in the Tibetan Plateau after fencing has been reported due to the colonization capacity of the vegetation (Shang et al., 2013) and the prevention of livestock herbivory on forage grasses, especially for graminoids and sedgy species that are palatable to livestock (Wu et al., 2009).

Species diversity, indicated by the Pielou evenness index, Shannon–Wiener diversity index, and Simpson dominance index, showed no statistically significant difference between GE plots and FG plots (Table 2). Similar results had also been reported in the steppe rangelands of the Central Anatolian Region in Turkey (Firincioğlu, Seefeldt & Şahin, 2007) and in the temperate semidesert rangelands of Nevada in North America (Courtois, Perryman & Hussein, 2004). However, the negative consequences for biodiversity after long-term grazing exclusion have also been found in many types of grassland ecosystems (Schultz, Morgan & Lunt, 2011; Maccherini & Santi, 2012). Therefore, there is no general agreement about the species diversity response to grazing exclusion in grassland ecosystems. On one hand, changes in plant species diversity due to grazing or grazing exclusion depend on resource partitioning and competitive patterns in vegetation; for instance, some species with lower competitive ability are reduced in density or disappear from the plant community entirely because of competition, light resources or nutrient availability (Grime, 1998; Van der Wal et al., 2004). On the other hand, the biodiversity response also depends on regional variation in major habitat characteristics, such as soil fertility, soil water availability, and growing-season precipitation (Olff & Ritchie, 1998; Wu et al., 2012; Wu et al., 2014).

A comparison of the community characteristics among the three alpine grassland types showed that total cover, Simpson index, Shannon index, and Pielou index were not significantly different between FG plots and GE plots in all three grassland types, except for that grazing exclusion resulted in the community vegetation heights increasing by 3.01 cm and 2.74 cm in alpine meadow and alpine steppe (P < 0.05), respectively (Table 2). Furthermore, statistical analyses showed that grazing exclusion had a significant effect on vegetation total cover and vegetation height, but grazing exclusion did not affect biodiversity indices (Table 3). This result indicated that short-term grazing exclusion resulted in increase of vegetation growth, but did not lead to obvious change in community composition in degraded alpine grassland ecosystems. The main differences of the plant community characteristics mainly come from the growing season climate differences of alpine grasslands (Table 3).

Impacts of grazing exclusion on biomass

Biomass is often considered a good approximation of productivity, especially in grassland communities (Chiarucci et al., 1999). The aboveground biomass of the GE plots was 31.82% higher than those of the FG plots (P < 0.05, Table 2). Therefore, the grazing exclusion resulted in obvious improvements in community aboveground biomass of degraded alpine grassland. Previous studies found that grazing exclusion significantly increased the total above-ground biomass of alpine meadows in the Tibetan Plateau, and in the fenced meadow, four functional groups, including the grass species group, the sedge species group, the leguminous species group and the noxious species group showed an increase in biomass, whereas only the forbs species group showed a decrease (Zhou et al., 2006; Wu et al., 2009). Wu et al. (2013) found that grazing exclusion increased total aboveground biomass by 27.09% in the Changtang region of Tibet. Their results are strongly consistent with the results of this study. The distinct and positive effect of grazing exclusion on biomass is mainly attributed to the absence of disturbance from herbivorous livestock (Mata-González et al., 2007; Wu et al., 2009); it may secondarily be attributed to the improvement of soil conditions (soil organic carbon and nitrogen storage, water infiltration rate, basal soil respiration, temperature, and moisture) after grazing exclusion, which favours the regeneration and the development of herbaceous species (Zhao et al., 2011; Mofidi et al., 2013).

Among three alpine grassland types, for alpine meadows, the biomass indices, including the mean values of the aboveground, belowground, and total biomass, tended to be higher in GE plots compared to FG plots, but the difference between then were not statistical significant. Nevertheless, for alpine steppe, the aboveground, belowground, and total biomass were all significantly higher due to grazing exclusion; For alpine desert steppe, only the aboveground biomass was significantly higher in fenced plots. Wu et al. (2013) also investigated the effect of grazing exclusion on alpine grasslands in the same region in Tibet, and found that grazing exclusion tended to increase aboveground biomass 17.80% in alpine meadow, 34.78% in alpine steppe, and 12.99% in alpine desert steppe, respectively; although these biomass values were also not statistical significant different from those of free grazed grasslands. However, grazing exclusion resulted in the improvement of aboveground biomass in whole alpine grasslands (alpine meadow + alpine steppe + alpine desert steppe) across regional scale in Tibet through the results from both Wu et al. (2013) and our study (Table 2).

There are increasing evidences show that precipitation plays a key role in the spatial distribution of species richness and diversity, primary production, and carbon and water cycles of alpine grassland ecosystems in this region (Hu et al., 2010; Yang et al., 2010; Wu et al., 2012; Wu et al., 2013). The precipitation gradients distributions control vegetation growth and community composition were also found in the present study which GSP had significant effect on all biomass indices of alpine grasslands, as well as relative community characteristic indices (Table 3). In fact, the similar results were also reported by Wu et al. (2012) and Wu et al. (2014)) in this region, therefore, these potential shift of GSP in Tibet should be considered when recommending any policies designed for the vegetation restoration of degraded alpine grasslands in the future.

The values of the aboveground, belowground, and total biomass were positively correlated with total vegetation cover in the alpine grasslands of the Tibet (Table 4). In addition, the total vegetation cover of alpine grasslands increased after continuous grazing exclusion (Table 2). Therefore, it is suggested that the higher biomass in GE plots was due to the increased vegetation cover. Other studies demonstrated that the grassland biomass and vegetation cover could simultaneously decrease or increase with grazing or not (Gao et al., 2009; Li et al., 2011). The biomass and vegetation cover simultaneously increased due to grazing exclusion was because of the absence of disturbance from herbivorous livestock (Jeddi & Chaieb, 2010), and also because of changes in plant competition and reproduction (Jing, Cheng & Chen, 2013). Moreover, the higher values of aboveground biomass and coverage of certain dominant species in communities under grazing exclusion would result in changes in species’ dominance and community composition (Wu et al., 2013). These results were partially validated and expanded upon in our study which the species diversity slightly declined in GE plots with the increasing of grassland biomass and vegetation cover (Table 2). Furthermore, the vegetation cover was negatively correlated with plant biodiversity indicators, D, H and E (P < 0.01) and the aboveground biomass of alpine grassland was negatively correlated with E (P < 0.01) (Table 4).

Conclusions

The restoration of degraded grassland ecosystem is a complex and long-term ecological process (Gao et al., 2014; Jing et al., 2014). Six to eight years of grazing exclusion in Tibet has not changed species diversity as indicated by the Pielou evenness index, Shannon–Wiener diversity index, and Simpson dominance index, but has significantly improved total vegetation cover, the vegetation height of community and the aboveground biomass of degraded alpine grasslands. These results demonstrate that grazing exclusion is an effective measure for maintaining community stability and improving aboveground vegetation growth in alpine grasslands. Nevertheless, it is worth mentioning that from the ANCOVA analysis, the growing season precipitation (GSP) had a significant effect on all vegetation indicators, except for vegetation height; but grazing exclusion only significantly affected vegetation cover, vegetation height and aboveground biomass (Table 3). Therefore, the GSP plays a more important role than grazing exclusion in which influence on plant community characteristics and biomass in alpine grasslands. In addition, the improvement of the vegetation cover, height and aboveground biomass due to the absence of disturbance from herbivorous livestock in the present study come from the examination short-term (6–8 years) effects of grazing exclusion, so it is questionable whether these improvements will be continuable if grazing exclusion is continuously implemented. Long term observations may be necessary to assess the ecological effects of the grazing exclusion management strategy on degraded alpine grasslands in Tibet. Thus, there is a need for continued research on the role of fencing on grassland restoration, management, and utilization in future.

Supplemental Information

Supplemental Information 1 Raw data of plant community characteristics and biomass

Values of plant community characteristics and biomass in three alpine grassland types at nine sampling sites under free grazing and grazing exclusion conditions.

Click here for additional data file.

Additional Information and Declarations

Competing Interests

Author Contributions

The authors declare there are no competing interests.

Yan Yan performed the experiments, wrote the paper, reviewed drafts of the paper.

Xuyang Lu conceived and designed the experiments, analyzed the data, contributed reagents/materials/analysis tools, prepared figures and/or tables, reviewed drafts of the paper.

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
