# Peer review of "Is grazing exclusion effective in restoring vegetation in degraded alpine grasslands in Tibet, China?"

_PeerJ, doi:10.7717/peerj.1020_

## Round 0.1 · original submission · Major Revisions

Please address major concerns about the methods as pointed out by both reviewers.

·

Basic reporting

No comments

Experimental design

No Comments

Validity of the findings

No comments

Additional comments

General comments
The study had showed the result of short time grazing bar on alpine grassland, which had very interesting point to guideline the grassland management in Tibet. The paper just showed the vegetation result, which just a part of restoration of degraded grassland, and should included the soil system. However, the paper had very good value to discuss the grazing exclusion’ impact on Tibetan plateau, which is the important data source for data gathering in the point of grazing system. Then I recommended to accept this paper after a minor revision.
Specific comments
1. The paper need to clear the method of grazing exclusion of whole year or just seasonal exclusion?
2. In the introduction, you should state the very highlight gap of your study and the past progress of grazing exclusion in Tibetan plateau.
3. Some background information of grazing system, grazing pressure, animal kind of grazing, etc, should be present in the method part.
4. Biomass is the total biomass of all plant root? Live root, died root?
5. Line 159: you should make a very clear of the ‘ni’ in the equation, why you did use the biomass as the Pi?
6. Line 160: the ’total observed number of species in the community’, please define the ‘community’ , the area? The scale? Because of the diversity index should be based on the ‘area’.
7. You did the different grassland type, to show the grazing exclusion result, however, is reasonable of Two-way ANOVA, which included the grassland type? Becaue of the grassland type is, in you study, from different county, and which is different definitely. Then, maybe, I think you can use One-way ANOVA.
8. Some new reference should be cited, such as ‘The sustainable development of grassland-livestock systems on the Tibetan plateau: problems, strategies and prospects, 2014, Rangeland Journal’, ‘Enhancing the Resilience of Coupled Human and Natural Systems of Alpine Rangelands on the Qinghai-Tibetan Plateau, The Rangeland Journal, Volume 37 2015’.
9. A small suggestion for analysis the result which depend you. ‘ you can analysis the relationship between the restoration’s effect and site’s Altitude, maybe it is interesting. This suggestion just up to your interesting to do or not.

Reviewer 2 ·

Basic reporting

The grazing exclusion by fencing has been widely used as a mainly restoration measure of alpine grassland in Qinghai-Tibetan Plateau, and there are lack of evaluation of ecological benefits from grazing exclusion management in recent years, so this manuscript is an interesting study that is a detail investigation of the effects of grazing exclusion on alpine grassland ecosystem in Qinghai-Tibetan Plateau. However, it has some shortcomings that should be corrected to improve the value to the reader. In particular, the methodology still needs some explanation if the majority of readers are to find value in the paper.

Experimental design

More detail of field measurements is required in the text on how this was done. Please clarify why used 0.5m*0.5m quadrats in different alpine grassland types? Which belowground plate components? (L134);

Validity of the findings

No comments

Additional comments

1. Author should discuss the impacts of climatic condition on the grazing exclusion effects on alpine grassland ecosystem.
2. In the Table 2, please indicate what data was cited from Wu et al (2013)?

---

## Round 0.2 · accepted · Accept

The revised version incorporates the suggestions by two reviewers and it improves the overall clarity of the paper.